# Leishmanicidal and healing effects of 3β,6β,16β-trihydroxy lup-20 (29)-ene isolated from *Combretum leprosum* on *Leishmania braziliensis* infection *in vitro* and *in vivo*

**Aline Sombra Santos[1], Naya Lúcia de Castro Rodrigues[1], Francisco Rafael Marciano Fonseca[1,2], Nathalia Braga Fayão Oliveira[1], Bianca Oliveira Loucard[2], Fabíola Fernandes Heredia[1], Teresa Neuma Albuquerque Gomes Nogueira[1], Ticiana Monteiro Abreu[1], Hélcio Silva dos Santos[3], Edson Holanda Teixeira[4], Luzia Kalyne Almeida Moreira Leal[2], Regis Bernardo Brandim Gomes[5], Clarissa Romero Teixeira[5], Maria Jania Teixeira[1] ***

1 Parasitology Laboratory, Department of Pathology and Legal Medicine, Faculty of Medicine, Federal University of Ceará, Fortaleza, Ceará, Brazil, 2 Laboratory of Pharmacognosy and Pharmaceutical Technology, Department of Pharmacy, Faculty of Pharmacy Dentistry and Nursing, Federal University of Ceará, Fortaleza, Ceará, Brazil, 3 Universidade Estadual Vale do Acaraú, Sobral, Ceará, Brazil, 4 Integrated Laboratory of Biomolecules, Department of Pathology and Legal Medicine, Faculty of Medicine, Federal University of Ceará, Fortaleza, Ceará, Brazil, 5 FIOCRUZ Ceará, Oswaldo Cruz Foundation, Eusébio, Fortaleza, Ceará, Brazil

* mjteixeira601@gmail.com

## Abstract

Treatment of cutaneous leishmaniasis depends on drugs that potentially cause serious side effects and resistance. Thus, topical therapies are attractive alternatives to the drugs currently used. 3β, 6β, 16β-trihydroxylup-20 (29)-ene is a lupane triterpene isolated from *Combretum leprosum* Mart. leaves (CLF-1), with reports of *in vitro* antileishmanial effect against *L. amazonensis* and to promote lesion healing in animal model. Herein, we evaluated the *in vitro* and *in vivo* antileishmanial and healing effects of CLF-1 against *L. braziliensis*. CLF-1 treatment showed low toxicity in macrophages and significantly reduced parasite load *in vitro*. CLF-1 induced higher IL-12 and TNF-α production and more discrete IL-4 and IL-10 production. For *in vivo* evaluation, a CLF-1 cream formulation was prepared to treat hamsters infected with *L. braziliensis*. CLF-1 treatment was able to reduce parasite load of the infected skin and lymph node more efficiently than the conventional treatment. Histopathological analysis indicated a strong inflammatory response accompanied by an important healing response. Data from this study indicate that topical CLF-1 treatment was effective and non-toxic in *L. braziliensis* infected hamsters suggesting its potential for further development as a future therapeutic intervention.

## Introduction

Leishmaniasis are parasitic diseases typical of tropical and subtropical regions, present in 102 countries. In Brazil, studies report the occurrence of about 20,000 new cases of the disease per year [1].

**Data Availability Statement:** Data are available from Zenodo ((DOI: 10.5281/zenodo.8064400).

**Funding:** The author(s) received no specific funding for this work.

**Competing interests:** The authors have declared that no competing interests exist.

The disease can manifest in visceral or tegumentary forms and the clinical spectrum depends on complex interactions, including parasite and host, such as *Leishmania* species and tropism, and the host immune status [2]. In Ceará, Brazil, cutaneous leishmaniasis (CL) has as main etiological agent *L. braziliensis* [3], a species capable of inducing localized cutaneous leishmaniasis, as well as mucocutaneous involvement with mucosal lesions of the oropharynx, and high morbidity [4].

The vertebrate host's main defense against leishmaniasis is through cell-mediated immune response, demonstrated by lymphocyte proliferation and production of the inflammatory cytokines IFN-γ and TNF-α [5]. However, these cytokines also mediate tissue damage, high levels of IFN-γ and TNF-α have been associated with increased inflammatory reaction and development of skin ulcers and mucosal lesions [6]. In *L. braziliensis* infection in humans, it has been shown that the balance between IFN-γ and IL-10 is crucial for wound healing [7]. IL-10 has an anti-inflammatory effect, counteracting the inflammatory effect of IFN-γ and TNF-α [8, 9].

The first-choice treatment for leishmaniasis has been done for over 5 decades with pentavalent antimonial. The toxicity of these agents and the persistence of side effects, even after dose level modification and treatment duration, are the main reported drawbacks [10]. Alternative treatment, such as liposomal amphotericin B and pentamidine, also have several side effects and inconveniences, including the high cost [11, 12]. In addition, the development of current drug resistant *Leishmania* strains in recent years has made the disease increasingly difficult to treat [13].

In the absence of a vaccine, there is an urgent need for more effective drugs to replace or supplement commonly used drugs. Natural herbal products are potential valuable sources for new medicinal agents. Several classes of plant-extracted natural products have shown promising leishmanicidal potential *in vitro* and/or *in vivo*, such as flavonoids, coumarins, quinones, quinoline alkaloids and terpenes, among others [14]. Lupane triterpene, 3β,6β,16β-trihydroxilup-20(29)-ene, isolated from *Combretum leprosum* Mart. (CLF-1) exhibited leishmanicidal activity against *L. amazonensis* promastigotes and amastigotes *in vitro* [15, 16]. In addition, CLF-1 demonstrated healing effect in a murine skin wound model [17].

To contribute to these studies, we investigated the leishmanicidal, inflammatory and healing effects of CLF-1, a natural product of the triterpene class, isolated from the leaves of a Brazilian northeast plant, *Combretum leprosum* Mart., *in vitro* and *in vivo* against *L. braziliensis* infection. Additionally, a cream formulation of CLF-1 for topical treatment was prepared and tested resulting in enhanced lesion healing. The result of this work suggests that CLF-1 could be a safe, effective alternative treatment option for control of leishmaniasis.

## Material and methods

### Triterpene 3β,6β,16β-triidroxilup-20(29)-ene: CLF-1

Leaves of *Combretum leprosum* were collected in June 2009 at Salgado dos Machados district, located 15 km from the city of Sobral, Ceará, Brazil. The plant classification was performed by a plant taxonomist from Acaraú Valley State University (Sobral, Brazil). A specimen of this plant was deposited in Herbarium Francisco José de Abreu Matos (Sobral, Brazil), under N∘ 4573. As previously described, the 3β,6β,16β-trihydroxy lup-20(29)-ene (CLF-1) was obtained initially by extraction from fresh *C. leprosum* leaves with EtOH/$H_2O$ (v/v, 8:2) for 15 days, filtered and evaporated under reduced pressure. The extract was fractioned by liquid chromatography over silica gel to purify CLF-1 (Fig 1). For structural analysis infrared spectra were recorded using a Perkin-Elmer 1000 spectrophotometer and $^1$H and $^{13}$C NMR were recorded on a Bruker Avance DPX-500 [18]. For *in vitro* experiments, a stock solution of CLF-1 was

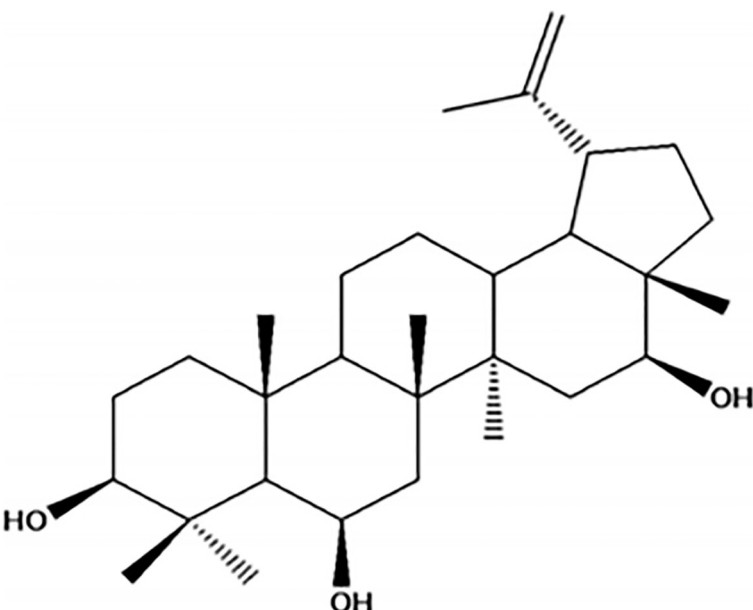

**Fig 1. CLF-1 or 3β,6β,16β-trihydroxy lup-20(29)-ene chemical structure isolated from *C. leprosum* leaves alcoholic extract [16].**

prepared freshly prior to use, using dimethylsulfoxide (DMSO) as a solvent. The appropriate concentrations were obtained by diluting the stock solution in sterile distilled water, and the final concentration of DMSO in the incubation mixture never exceeded 0.1%. Control samples were always treated with the same amount of DMSO (0.1% v/v) as used in the corresponding experiments.

## Parasites

*Leishmania (Viannia) braziliensis* strain (MCAN/BR/98/R619) was kindly provided by Prof. Dr. Maria Alda da Cruz from Oswaldo Cruz Foundation (FIOCRUZ-RJ) and were kept frozen at -80˚C at the Parasitology Laboratory of the Federal University of Ceará. Parasites were cultured at 25˚C in N.N.N. medium containing Schneider supplemented with 20% inactivated fetal bovine serum (SBF), 2% sterile human urine, 100 U/mL penicillin and 100 μg/mL streptomycin.

## Animals

Female Syrian hamsters (*Mesocricetus auratus*), 6 to 8 weeks old, weighing around 80 to 90g, were obtained from the central animal facility of Pathology and Legal Medicine Department of Federal University of Ceará (DPML/UFC), maintained at 25˚C, in an alternating light/dark cycle, with food and water *ad libitum*. The Animal Care and Utilization Committee from UFC approved all experimental procedures (Protocol n°· 6490040418).

## CLF-1 cream

A cream formulation containing 50 μg of CLF-1/15 mg base cream was used for *in vivo* treatment. The cream was formulated with cetearyl alcohol with Steareth-10 and -20 (5%), isopropyl palmitate (5%), butylated hydroxytoluene (BHT) (0.06%), propylparaben (0.06%),

methylparaben (0.18%), glycerin (5%), disodium EDTA (0.12%), distilled water and 1% EDTA 2020 gel base (15%) in the Pharmacognosy and Pharmaceutical Technology Laboratory (Faculty of Pharmacy—UFC). As a control, the vehicle (1% EDT 2020 base gel) was used.

## Effect of CLF-1 on promastigotes

Culture containing promastigote forms were centrifuged at 3.000 rpm for 15 minutes at 4˚C, resuspended in Schneider medium, counted, and diluted in supplemented Schneider medium to obtain a concentration of $1x10^7$ promastigotes/mL. Promastigotes were distributed in 48-well plates in a volume of 160 μL in each well and diluted CLF-1 was added at the respective concentrations of 5, 10, 25, 50 and 100 μg/mL. For control, 1% DMSO, Amphotericin B (16 μg/mL) and non-supplemented Schneider medium were used [19]. The plates were incubated for 24 and 48 hours at 25˚C and the number of viable promastigotes was determined.

## Cytotoxicity assay in macrophages

J774 macrophages were cultured in supplemented RPMI medium and maintained at 5% $CO_2$, 95% humidity at 37˚C. J774 macrophages (5 x $10^5$ cells/mL) were incubated with CLF-1 (5, 10, 25, 50 and 100 μg/mL), DMSO 1% (CLF-1 vehicle), DMSO 10% (cytotoxic standard) and Amphotericin B in 96-well microplates. Cytotoxicity in J774 macrophages was evaluated after 24 and 48 h using the standard MTT colorimetric assay. For reading, each plate was shaken for 10 minutes, and absorbance was measured on a spectrophotometer with a wavelength of 570 nm.

## Effect against intracellular amastigotes

J774 macrophages ($1x10^6$ macrophages/m) were distributed in 24-well flat-bottomed plates containing round glass coverslips (23 mm). For macrophage infection with *L. braziliensis*, parasites were added at a concentration of $1x10^7$ parasites/mL. The infected macrophages were incubated for an additional 24 h and then CLF-1 (10, 25, 50 and 100 μg / mL), 1% DMSO, Glucantime® (4 mg/mL) and Interferon-γ (20 ng/mL) were added. Plates were incubated at 5% $CO_2$ at 37˚C and 95% humidity for 24 and 48 hours. After this time, the supernatants were removed from the wells and then stored at -20˚C for further cytokine analysis. For parasitic load evaluation, coverslips were removed from each well and washed with saline, fixed, and stained with Giemsa and examined under optical microscope. Amastigotes were counted in 50 typical macrophages on each coverslip.

## Evaluation of cytokine production

Production of IL-10, IL-4, IL-12 and TNF-α cytokines was determined by ELISA using supernatants obtained from culture of *L. braziliensis*-infected macrophages. Assays were performed according to the instructions of the ELISA kit manufacturer (BD Biosciences).

## *In vivo* infection and treatment with CLF-1 cream

Stationary phase *L. braziliensis* promastigotes were inoculated into the right ear dermis of each animal, according to the groups, at a concentration of $1x10^7$ in 20 μL of sterile saline [20]. Animals (n = 30) were randomly separated into 3 groups (10 animals each) and anesthetized with ketamine (80 mg/kg, i.p.) and xylazine (15 mg/Kg) before infection: 1. Group infected and treated with the cream vehicle (negative control); 2. Group infected and treated with Glucantime® (treatment control group, 100 mg/Kg, IM); 3. Group infected and treated with CLF-1 cream formulation (50 μg/ 15mg). Treatment began shortly after the onset of the ulcerated

lesion 15 days after infection and was performed for 10 consecutive days. Glucantime® was administered intramuscularly (100 mg/kg, I.M., 80 μL in each animal), alternating between right and left thigh each day. The amount of cream containing or not CLF-1 was measured on a precision scale using a flat spatula containing the cream. The ear lesion was exposed with the help of tweezers and the ointment was applied over the lesion. Lesion thickness of the right and left ears (control) was measured for 45 days with a circular caliper (Mitutoyo, Japan). Lesion thickness was assessed as the difference between the inoculated ear and the uninoculated collateral ear.

## Parasitic load evaluation

Animals were euthanized with an i.p. injection of ketamine (600 mg/kg) and xylazine (30 mg/kg) after 10 days of treatment, 25 days after infection. Infected ears and retromaxillary lymph nodes were collected for further studies of parasitic burden and histopathological analysis. To quantify the number of parasites in the retromaxillary lymph nodes and infected ears the limiting dilution technique was used [21]. Plates were sealed and incubated at 25˚C and observed under an inverted microscope every 3 days for up to 30 days to record parasite dilutions. The plate reading result was recorded in the ELIDA 12c software for Windows for the final calculation of the number of parasites present in the samples [22].

## Histopathological analysis

For analysis of the histopathological aspects, infected and uninfected ears of each group were removed and fixed in 10% buffered formalin, processed, and stained with hematoxylin-eosin. The changes were analyzed under a standard optical microscope.

## Statistical analysis

To verify the statistical significance between the treated groups and the control, the normality test was applied and when there was normal distribution, the t-Student test was used. For comparisons between multiple groups, the one-way ANOVA test was performed, followed by the Bonferroni post-test. The tests were performed using the GraphPad Prism Program version 7.0. In all tests used, the minimum accepted significance was $P < 0.05$.

## Results

### Cytotoxic effect of CLF-1 *in vitro*

Initially, we characterized the effect of J774 macrophages and *L. braziliensis* promastigotes cultivated in the presence of different concentrations of CLF-1 for 48 hours. Lower CLF-1 concentrations such as 5, 10, and 25 μg/mL were less cytotoxic at levels very similar to the culture medium and drug vehicle (Fig 2A and 2B). CLF-1 presented a significant cytotoxic effect on macrophages only at the highest concentration tested (100 μg/mL) (Fig 2A and 2B). The leishmanicidal effect of CLF-1 on promastigotes was observed in a dose dependent manner (25, 50 and 100 μg/mL) significantly decreasing the number of viable promastigotes compared to medium containing only promastigotes at 24 sustained at 48 hours (Fig 2C and 2D). CLF-1 concentrations of 50 and 100 μg/mL showed no differences in percent promastigote viability compared to amphotericin B at 24 hours (Fig 2A and 2B).

### CLF-1 treatment control *L. braziliensis* infection inside macrophages

The antileishmanial effect of CLF-1 was tested in intracellular amastigotes of *L. braziliensis*. Interestingly, CLF-1 treatment was able to significantly reduce the number of amastigotes at a

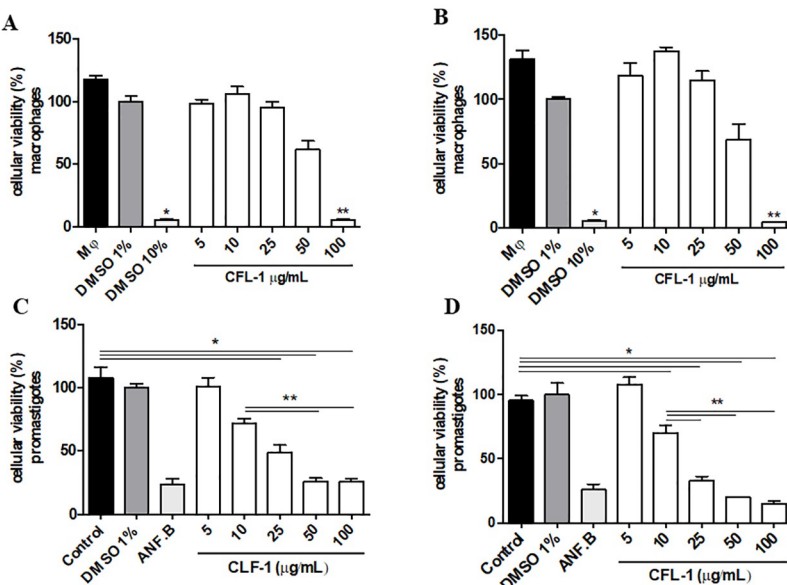

**Fig 2.** Effect of CLF-1 treatment on cell viability *in vitro* of J774 macrophages treated with CLF-1 for 24 (A) and 48 (B) hours and *Leismania. braziliensis* promastigotes for 24 (C) and 48 hours (D). Control: *L. braziliensis* promastigotes or J774 macrophages without any treatment; DMSO: dimetilsulfoxide; ANF.B: Amphotericin B; Mφ: macrophages. p < 0,0001.

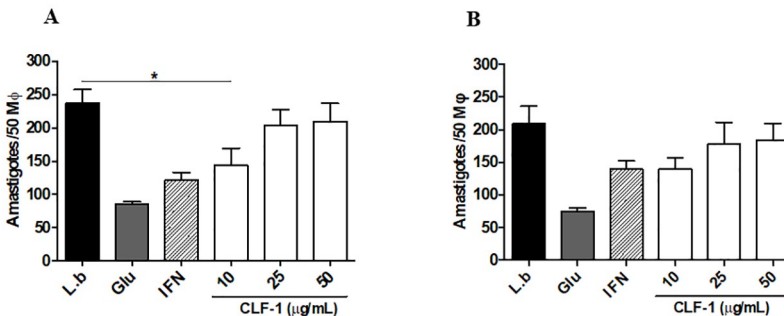

**Fig 3. Effect of CLF-1 treatment on *Leishmania braziliensis* infected macrophages.** (A) Parasite load after treatment with CLF-1 for 24 and (B) 48 hours. L.b.: *Leishmania braziliensis* infected macrophages; Glu: Glucantime; IFN: Interferon-gamma (p < 0,0001).

lower concentration (10 µg/mL). This effect was not observed at higher CLF-1 concentrations (Fig 3A and 3B).

### Effect of CLF-1 treatment modulate cytokine production by *L. braziliensis*-infected macrophages

Next, we analyzed the profile of cytokine production by infected macrophages following CLF-1 treatment. IL-12 production was significantly increased following treatment with CLF-1 up to 48 hours (Fig 4A and 4B). A significant production of TNF-α was also detected at 24 hours that decreased at 48 hours. Interestingly, CLF-1 treatment also induced production of anti-inflammatory cytokines, IL-4, and IL-10 at 24 and 48 hours (Fig 4A and 4B).

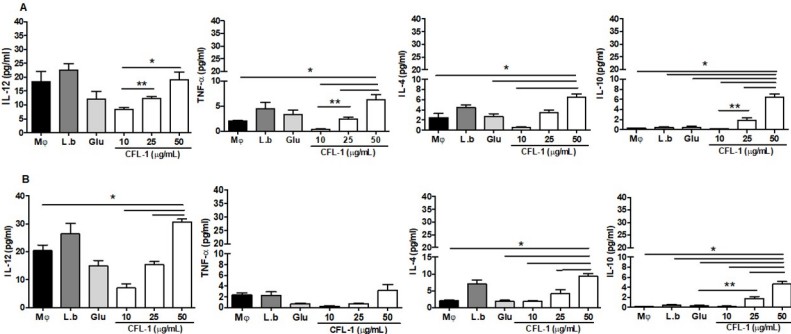

**Fig 4. Production of cytokines *in vitro* by *Leishmania braziliensis* infected macrophages treated with CLF-1 for 24 (A) and 48 (B) hours.** *L. braziliensis* infected macrophages were treated with CLF-1 and IL-12, TNF-α, IL-4 and IL-10 production were measured in the supernatant. MØ: macrophages; L.b: *Leishmania braziliensis* infected macrophages; Glu: Glucantime; p< 0,0001.

## CLF-1 treatment of hamsters infected with *L. braziliensis* induced lesion healing and reduced parasite load

We extended our investigation to *in vivo L. braziliensis* infection using the hamster model. Overall, topical treatment with CLF-1 was performed for ten consecutive days after the onset of the ulcerated lesion (15 days after infection) resulting in decreased ear thickness that coincided with lesion healing (Fig 5A and 5C). Treatment with glucantime had few animals with ulcerated lesions and were able to reduce lesion thickness earlier but a significant reduction was only observed after treatment conclusion. Although lesions appeared in all animals, it was possible to observe some degree of variation. A significant difference between lesions could be observed at the last day of treatment when CLF-1 and glucantime treated groups recovered faster than the control group that presented lesions that were visually larger and ulcerated (Fig 5B).

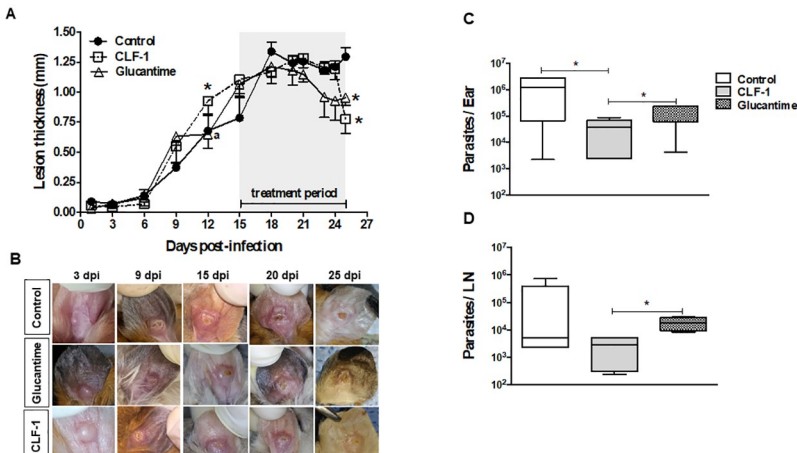

**Fig 5. Effect of topical CLF-1 treatment in hamsters infected with *L. braziliensis*.** Lesion development in infected hamsters treated for 10 consecutive days with topical CLF-1. (A) Lesion thickness and (B) Representative pictures of lesions from infected ears on the last day of treatment. Parasite load determination in the ear (C) and draining lymph node (D) from hamsters infected with *L. braziliensis* and treated with CLF-1 dpi = days post infection; ** and * = significance in relation to control. *p < 0,05.

Moreover, animas treated with CLF-1 lesions did not present ulcerated lesions with a necrotic center and there was a significant parasite load reduction at the lesion and lymph node comparing to the group treated with glucantime (Fig 5C and 5D).

## Histopathological analysis of infected ears from hamsters treated with CLF-1

The main histopathological finding was the presence of an intense inflammatory exudate with granulomas, especially lesions that still had ulcers on the last day of treatment, regardless of the group observed. The inflammatory infiltrate was mainly composed by macrophages, neutrophils, and lymphocytes (Fig 6B). We observed an extensive presence of apoptotic bodies and small vessels formed in the area closest to the ulcer. The lesion of control animals presented areas of fibrinoid necrosis in addition to an inflammatory exudate composed by infected macrophages. The group treated with antimony presented animals that had non-ulcerated tissue (Fig 6C and 6E). Apoptotic bodies were present with greater intensity and granulomas were present in a more advanced stage in the group treated with CLF-1, indicating a possible control of parasite proliferation (Fig 6F). Treatment with CLF-1 showed a more extensive and more robust re-epithelialization and neovascularization, a hallmark for the healing process (Fig 6D). Therefore, treatment with CLF-1 resulted in an inflammatory response and strong signs of healing with re-epithelialization of the lesion area.

## Discussion

Development of new drugs for CL treatment that are less toxic, more effective and that could be topically or orally administered not requiring outpatient support is critical [23, 24]. *L. braziliensis* is the major species related to American tegumentary leishmaniasis and treatment is recommended to prevent spread and disfiguring lesions but limited to a few drugs that have very toxic side effects and parasite resistance.

CLF-1 is isolated from *Combretum leprosum* Mart., a native species that is commonly used as a healing agent to treat skin diseases using plasters to cover wounds with leaves that contain a high concentration of CLF-1 [17]. Previous results demonstrate that CLF-1 presents anti-eishmanial activity against *L. amazonensis in vitro* [16]. Here we further explore the anti-leishmanial potential of CLF-1 using the hamster model of *L. braziliensis* infection employing a topical CLF-1 cream formulation. The use of topical formulations poses advantages such as easy application, with low toxicity and adverse effects, increasing chances of treatment compliance and success. The effectiveness of local therapies has been reported previously. Topical treatment using Arnica tincture showed antileishmanial activity in infected hamsters [25]. A pentamidine cream and a topical miltefosine gel was also tested in *L. braziliensis* infected mice [26, 27].

Importantly, a CLF-1 topical formulation was previously tested demonstrating a healing effect on skin lesions in rats [17]. Treatment of infected golden hamsters with CLF-1 was performed for ten days topically starting at lesion onset, the time when most patients seek medical treatment. The antileishmanial effect of CLF-1 induced a decrease in parasite load, promoting lesion healing with lesion thickness reduction at the end of treatment. In control mice, dermal lesions were visually larger and histopathological examination showed the presence of an intense inflammatory infiltrate, containing epidermal hyperplasia and ulcerated areas, characteristic of CL lesions. CLF-1 healing effect was also observed in the histopathological analysis of the infected ears, showing reepithelialization and the presence of collagen fibers. Healing coincide with parasite load reduction, both in the lesion and draining lymph node from animals treated with CLF-1.

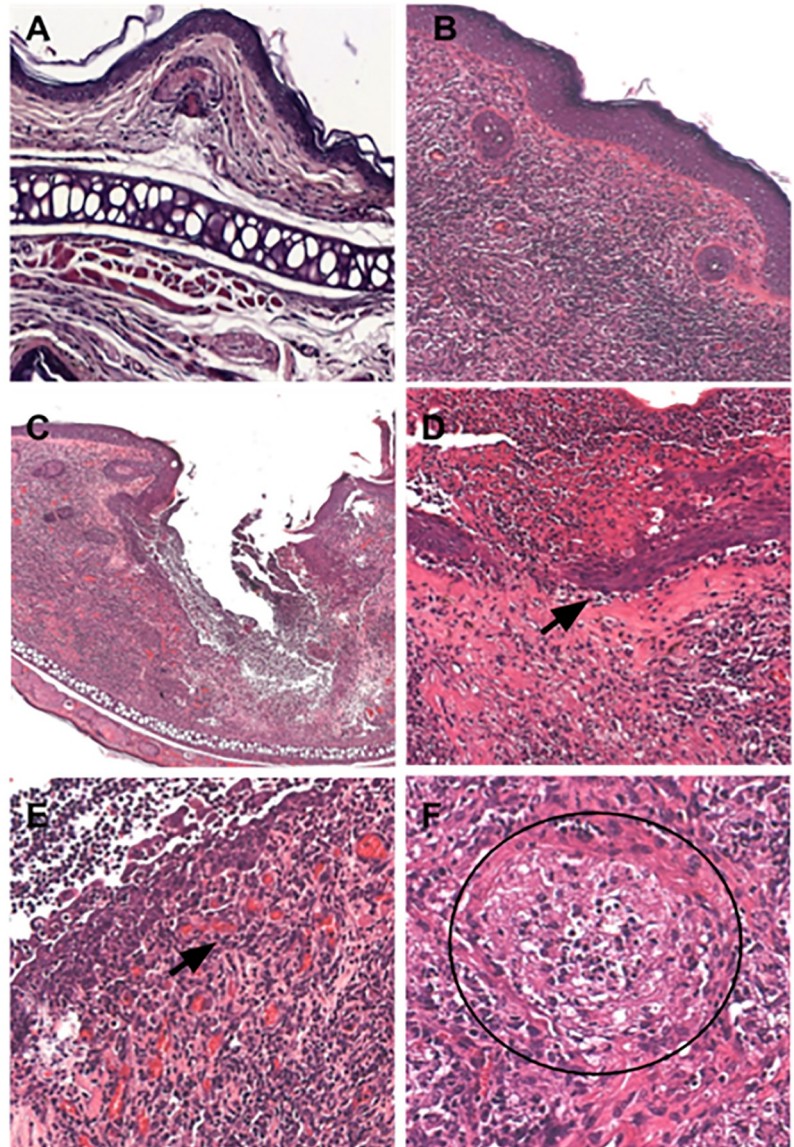

**Fig 6. Main histopathological findings of hamsters infected with *Leishmania braziliensis* and treated with CLF-1.**
(A) Uninfected ear (200x). (B) Untreated infected ear (200x). (C and E) Infected ear treated with Glucantime (100x and 400x, respectively). (D and F) Infected ear topically treated with CLF-1 (400x). Arrows indicate areas of neovascularization, black circle show an area with a granuloma with activated macrophages.

Cytokine production demonstrated that CLF-1 treatment of infected macrophages significantly increased IL-12 production accompanied by lower production of TNF-α, IL-4 and IL-10, indicating a mixed inflammatory environment. Although IL-12 and TNF-α are key cytokines that activate macrophages and promote *Leishmania* killing, excessive inflammation is related to the development of more severe clinical manifestations. Thus, the presence of IL-10 and IL-4 is also important, balancing the inflammatory response to prevent tissue damage [6, 28].

Although we have not explored CLF-1 anti-leishmanial mechanism, it was previously reported that CLF-1 is a potential inhibitor of topoisomerase IB [16]. Whether treatment CLF-

1 provides an additional effect against other forms of cutaneous leishmaniasis is a question that requires future investigation. Moreover, evaluation of combination therapy, topical CLF-1 and Glucantime, could increase treatment efficacy, lower drug usage, and reduce treatment length and side effects.

Together, the results of this study provide evidence of CLF-1 topical treatment of experimental CL in hamsters is an effective and promising treatment option to the chemotherapies currently available.

## Author Contributions

**Conceptualization:** Aline Sombra Santos, Ticiana Monteiro Abreu, Luzia Kalyne Almeida Moreira Leal, Maria Jania Teixeira.

**Data curation:** Aline Sombra Santos, Maria Jania Teixeira.

**Formal analysis:** Aline Sombra Santos, Naya Lúcia de Castro Rodrigues, Fabíola Fernandes Heredia, Teresa Neuma Albuquerque Gomes Nogueira, Ticiana Monteiro Abreu, Edson Holanda Teixeira, Luzia Kalyne Almeida Moreira Leal, Regis Bernardo Brandim Gomes, Clarissa Romero Teixeira, Maria Jania Teixeira.

**Investigation:** Aline Sombra Santos, Naya Lúcia de Castro Rodrigues, Francisco Rafael Marciano Fonseca, Nathalia Braga Fayão Oliveira, Bianca Oliveira Loucard, Fabíola Fernandes Heredia, Teresa Neuma Albuquerque Gomes Nogueira, Ticiana Monteiro Abreu.

**Methodology:** Francisco Rafael Marciano Fonseca, Nathalia Braga Fayão Oliveira, Bianca Oliveira Loucard, Fabíola Fernandes Heredia, Hélcio Silva dos Santos, Edson Holanda Teixeira, Luzia Kalyne Almeida Moreira Leal, Maria Jania Teixeira.

**Project administration:** Naya Lúcia de Castro Rodrigues, Fabíola Fernandes Heredia, Maria Jania Teixeira.

**Resources:** Bianca Oliveira Loucard, Hélcio Silva dos Santos, Edson Holanda Teixeira, Luzia Kalyne Almeida Moreira Leal.

**Supervision:** Maria Jania Teixeira.

**Writing – original draft:** Bianca Oliveira Loucard, Regis Bernardo Brandim Gomes, Clarissa Romero Teixeira, Maria Jania Teixeira.

**Writing – review & editing:** Regis Bernardo Brandim Gomes, Clarissa Romero Teixeira.

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
