## [Decision Letter · Decision Letter 0]

23 Mar 2023

PONE-D-23-02657Leishmanicidal and healing effects of 3β,6β,16β-trihydroxy lup-20 (29)-ene isolated from Combretum leprosum on Leishmania braziliensis infection in vitro and in vivo.PLOS ONE

Dear Dr. Teixeira,

Thank you for submitting your manuscript to PLOS ONE. After careful consideration, we feel that it has merit but does not fully meet PLOS ONE’s publication criteria as it currently stands. Therefore, we invite you to submit a revised version of the manuscript that addresses the points raised during the review process.

We look forward to receiving your revised manuscript.

Kind regards,

Manas Ranjan Dikhit

Academic Editor

PLOS ONE

Journal Requirements:

2. To comply with PLOS ONE submissions requirements, in your Methods section, please provide additional information on the animal research and ensure you have included details on (1) methods of sacrifice, (2) methods of anesthesia and/or analgesia, and (3) efforts to alleviate suffering.

Reviewers' comments:

Reviewer's Responses to Questions

**Comments to the Author**

1. Is the manuscript technically sound, and do the data support the conclusions?

Reviewer #1: Partly

Reviewer #2: Partly

2. Has the statistical analysis been performed appropriately and rigorously? 

Reviewer #1: Yes

Reviewer #2: Yes

3. Have the authors made all data underlying the findings in their manuscript fully available?

Reviewer #1: Yes

Reviewer #2: Yes

4. Is the manuscript presented in an intelligible fashion and written in standard English?

Reviewer #1: Yes

Reviewer #2: Yes

5. Review Comments to the Author

Reviewer #1: In this article by Santos et.al, entitled Leishmanicidal and healing……… on Leiahmania braziliensis

infection in vitro and in vivo” the authors have demonstrated anti-leishmanial and wound healing effect

of Combretum leprosum leaves extract. The article has important information and findings but requires

revision with clarification before being accepted for publication.

My comments are below.

1. Brief description of isolation and characterization of CLF-1 in page 5 line 106 should be provided.

2. The cream formulation should be elaborated in page 6 line 135.

3. In figure 1, panel A, B, C & D, the only macrophage bar in all the cases- it is not clear against

which control was these bars normalized. In panel ‘A’ the macrophage bar is around 120 %, in

panel ‘B’ it is around 130%, in ‘C’ it is 110% and in ‘D’ it is less than ‘100’. Please clarify and

correct. It is also unclear why 100µg/ml CFL-1 is toxic to macrophage but not to Leishmania in

that respect.

4. In figure 2 the authors should explain why there is not much effect of duration of treatment.

5. The data provided in the manuscript shows that higher dose of CFL-1 has no effect on number of

amastigotes (Fig 2) but higher dose of CFL-1 do induce the cytokines (Fig 3). How can this be

explained in wound healing and clearance of parasites.

6. I would suggest the authors to measure the parasitic load through some molecular method like

qPCR.

Reviewer #2: Author has shown with different experiments that CLF-1 has in vitro (in J774 macrophage) and in vivo (in Hamster model) anti-leishmanial effect against L. braziliensis. However, author should address the following comments before acceptance.

Comments:

1. Author did not study any mechanism of inhibition for CLF-1 in L. braziliensis as previous study already explored the mechanism of inhibition by CLF-1 in L. amazonensis promastigotes and also showed that CLF-1 has anti-leishmanial activity in-vitro. So, in my opinion the only novel part of this study was to explore the effect of CLF-1 cream in Hamster model. Author should study the mechanism of inhibition in amastigote by isolating the amastigotes from infected macrophages.

2. Author used J774 (mouse cancerous cell line) for in-vitro infection study. In my opinion, author may use mouse peritoneal macrophages for this study as this the primary macrophages.

3. For in-vitro study, author used Amphotericin B as a control and concentration used is 16 µg/ml. What is the rationale behind choosing this concentration? Is there any report?

4. Author has developed CLF-1 cream for topical application. However, author did not provide the details about the process of formulation of the cream.

5. Author has shown that promastigotes viability is reduced with increasing concentration of CLF-1 (Figure 1) and in case of intracellular amastigotes, viability is increased with increasing concentration of CLF-1 (Figure 2). What is the explanation of this contradictory results?

However, I am quite happy with the work but, author need to address the above issues.

6. PLOS authors have the option to publish the peer review history of their article (what does this mean?). If published, this will include your full peer review and any attached files.

Reviewer #1: No

Reviewer #2: No

---

## [Author Response · Author response to Decision Letter 0]

8 May 2023

ANSWERS TO REVIEWERS:

Reviewer #1: In this article by Santos et al, entitled Leishmanicidal and healing……… on Leishmania braziliensis infection in vitro and in vivo” the authors have demonstrated anti-leishmanial and wound healing effect of Combretum leprosum leaves extract. The article has important information and findings but requires revision with clarification before being accepted for publication.

My comments are below.

1. Brief description of isolation and characterization of CLF-1 in page 5 line 106 should be provided.

More information related to the isolation and characterization of CLF-1 was included in the Material and Methods section as suggested.

2. The cream formulation should be elaborated in page 6 line 135.

A more detailed description of CLF-1 cream formulation was included in the Material and Methods section as suggested.

3. In figure 1, panel A, B, C & D, the only macrophage bar in all the cases- it is not clear against which control was these bars normalized. In panel ‘A’ the macrophage bar is around 120 %, in panel ‘B’ it is around 130%, in ‘C’ it is 110% and in ‘D’ it is less than ‘100’. Please clarify and correct. It is also unclear why 100µg/ml CFL-1 is toxic to macrophage but not to Leishmania in that respect.

We appreciate that the reviewer’s comment and we have corrected figure 1, the bars were normalized based on the experiment controls. While CLF-1 cytotoxicity to macrophages was only observed with the highest concentration (100µg/ml), promastigotes were more susceptible to the toxic effect, observed also with lower CLF-1 concentrations (10 µg/ml). This is probably resulting from different mechanisms of actions of CLF-1 on such distinct cells (macrophages and Leishmania parasites) and should be further explored on future studies. 

4. In figure 2 the authors should explain why there is not much effect of duration of treatment.

We believe that we did not observe a significant effect on amastigotes at 24h, but not at 48h, probably due to the kinetics of CLF-1 effect on the intracellular parasites. Unfortunately, we were not able to test in vitro for a longer period. However, the CLF-1 anti-leishmanial effect was evident in vivo on infected hamsters where we observe a clear reduction of parasite numbers and lesions were not ulcerated with a necrotic center compared to the group treated with Glucantime.

5. The data provided in the manuscript shows that higher dose of CFL-1 has no effect on number of amastigotes (Fig 2) but higher dose of CFL-1 do induce the cytokines (Fig 3). How can this be explained in wound healing and clearance of parasites.

The mechanism of CLF-1 treatment on amastigotes (intracellular) and macrophages and the kinetics of the effect observed is probably different and it remains to be elucidated. CLF-1 has been previously described as having an antimicrobial, pro-healing and antiparasitic activities. The wound healing and clearance of parasites is probably resulting, at least in part, by direct modulation of cytokine production that results in parasite killing and promote lesion healing. 

6. I would suggest the authors to measure the parasitic load through some molecular method like qPCR.

Although qPCR is a sensitive and specific method, limiting dilution is more accessible, widely used, and a reliable method to quantify live parasites. Thus, we decided to measure the parasite burden using the limiting dilution method to determine the parasite load. This method has been used by our group in other studies (Dutra, et al., 2020 and Freire et al., 2022).

Reviewer #2: Author has shown with different experiments that CLF-1 has in vitro (in J774 macrophage) and in vivo (in Hamster model) anti-leishmanial effect against L. braziliensis. However, author should address the following comments before acceptance.

Comments:

1. Author did not study any mechanism of inhibition for CLF-1 in L. braziliensis as previous study already explored the mechanism of inhibition by CLF-1 in L. amazonensis promastigotes and also showed that CLF-1 has anti-leishmanial activity in-vitro. So, in my opinion the only novel part of this study was to explore the effect of CLF-1 cream in Hamster model. Author should study the mechanism of inhibition in amastigote by isolating the amastigotes from infected macrophages.

The reviewer made an important point about previous work demonstrating the potential protective and healing effect of CLF-1. Infection caused by Leishmania braziliensis and Leishmania amazonensis induce skin lesions, but disease progression and immune evasion mechanisms are distinct reinforcing the importance of testing CLF-1 against each individual Leishmania species. Investigation of the mechanism of action of CLF-1 treatment on amastigotes and macrophages are the subject of a future study. 

2. Author used J774 (mouse cancerous cell line) for in-vitro infection study. In my opinion, author may use mouse peritoneal macrophages for this study as this the primary macrophages.

Although we have not used a primary culture of macrophages due to a limitation on the number of experimental hamsters/mice available at the time, we believe that J774 cells, that are widely used for in vitro tests, also provide reliable and representative results.

3. For in-vitro study, author used Amphotericin B as a control and concentration used is 16 µg/ml. What is the rationale behind choosing this concentration? Is there any report?

We have used this concentration of Amphotericin B as a control for parasite killing. It was based on previous work published by the group (Pinheiro et al., 2021). This reference was included in the Material and Methods.

4. Author has developed CLF-1 cream for topical application. However, author did not provide the details about the process of formulation of the cream.

Details concerning the formulation of CLF-1 cream were included in the Material and Methods section as suggested.

5. Author has shown that promastigotes viability is reduced with increasing concentration of CLF-1 (Figure 1) and in case of intracellular amastigotes, viability is increased with increasing concentration of CLF-1 (Figure 2). What is the explanation of this contradictory results? However, I am quite happy with the work but, author need to address the above issues.

We observed a different effect on promastigotes and amastigotes probably as a result of the mechanism of action and kinetics of CLF-1 on the different stages (intracellular x extracellular). Unfortunately, we were not able to test in vitro for a longer period as we moved forward for in vivo experiments. The CLF-1 anti-leishmanial effect was evident in vivo on infected hamsters where we observe a clear reduction of parasite and lesions were not ulcerated with a necrotic center compared to the group treated with Glucantime.

---

## [Decision Letter · Decision Letter 1]

12 Jun 2023

Leishmanicidal and healing effects of 3β,6β,16β-trihydroxy lup-20 (29)-ene isolated from Combretum leprosum on Leishmania braziliensis infection in vitro and in vivo.

PONE-D-23-02657R1

Dear Dr. Maria,

We’re pleased to inform you that your manuscript has been judged scientifically suitable for publication and will be formally accepted for publication once it meets all outstanding technical requirements.

Kind regards,

Manas Ranjan Dikhit

Academic Editor

PLOS ONE

Additional Editor Comments (optional):

Reviewers' comments:

Reviewer's Responses to Questions

**Comments to the Author**

1. If the authors have adequately addressed your comments raised in a previous round of review and you feel that this manuscript is now acceptable for publication, you may indicate that here to bypass the “Comments to the Author” section, enter your conflict of interest statement in the “Confidential to Editor” section, and submit your "Accept" recommendation.

Reviewer #1: All comments have been addressed

2. Is the manuscript technically sound, and do the data support the conclusions?

Reviewer #1: Yes

3. Has the statistical analysis been performed appropriately and rigorously? 

Reviewer #1: Yes

4. Have the authors made all data underlying the findings in their manuscript fully available?

Reviewer #1: Yes

5. Is the manuscript presented in an intelligible fashion and written in standard English?

Reviewer #1: Yes

6. Review Comments to the Author

Reviewer #1: (No Response)

7. PLOS authors have the option to publish the peer review history of their article (what does this mean?). If published, this will include your full peer review and any attached files.

Reviewer #1: No

---

## [Editor Report · Acceptance letter]

10 Jul 2023

PONE-D-23-02657R1 

Leishmanicidal and healing effects of 3β,6β,16β-trihydroxy lup-20 (29)-ene isolated from *Combretum leprosum* on *Leishmania braziliensis* infection *in vitro* and *in vivo*

Dear Dr. Teixeira:

I'm pleased to inform you that your manuscript has been deemed suitable for publication in PLOS ONE. Congratulations! Your manuscript is now with our production department. 

Kind regards, 

on behalf of

Dr. Manas Ranjan Dikhit 

Academic Editor

PLOS ONE